# Intelligent Human–UAV Interaction System with Joint Cross-Validation over Action–Gesture Recognition and Scene Understanding

**Bo Chen [1,2,3,†], Chunsheng Hua [4,*,†], Decai Li [1,2], Yuqing He [1,2] and Jianda Han [5]** 

1   State Key Laboratory of Robotics, Shenyang Institute of Automation, Chinese Academy of Sciences, Shenyang 110000, China
2   Institutes for Robotics and Intelligent Manufacturing, Chinese Academy of Sciences, Shenyang 110000, China
3   University of Chinese Academy of Sciences, Beijing 100000, China
4   College of Information, Liaoning University, Shenyang 110000, China
5   Institute of Robotics and Automatic Information System, College of Artificial Intelligence, Nankai University, Tianjin 300000, China
*   Correspondence: huachunsheng@gmail.com
†   Both Hua and Chen contribute equally to this research and are the first authors.

**Abstract:** We propose an intelligent human–unmanned aerial vehicle (UAV) interaction system, in which, instead of using the conventional remote controller, the UAV flight actions are controlled by a deep learning-based action–gesture joint detection system. The Resnet-based scene-understanding algorithm is introduced into the proposed system to enable the UAV to adjust its flight strategy automatically, according to the flying conditions. Meanwhile, both the deep learning-based action detection and multi-feature cascade gesture recognition methods are employed by a cross-validation process to create the corresponding flight action. The effectiveness and efficiency of the proposed system are confirmed by its application to controlling the flight action of a real flying UAV for more than 3 h.

**Keywords:** action detection; gesture recognition; scene understanding; joint cross validation

## 1. Introduction

Following decades of efforts from both academic and industrial researchers, unmanned aerial vehicle (UAV) technology has made significant progress, and it has begun to play an increasingly important role in various fields, such as aerial photography, transportation and delivery, public surveillance [1], and modern agriculture. Consequently, the interaction between UAVs and humans has become more frequent, and a growing demand has emerged for additional smart human–UAV interaction and control [2]. That is because traditional human–UAV interaction is not easy, and requires training and concentration during the flight operation [3], which usually limits the application of UAVs to thoroughly trained and professional people [4].

To address this problem, numerous methods have been proposed, which can mainly be categorized as four types: wearable sensors [5], more user-friendly remote controllers (see Figure 1a,b), speech recognition [6], and gesture/action recognition [7–9]. Although the use of speech recognition appears to be the most efficient method for controlling the machine, and speech recognition has been applied in several language translation tasks, its performance will degrade significantly and become unreliable owing to factors such as background noise, and the wide variety of human voice accents and speaking speeds. Wearable sensors to control UAVs have also undergone rapid development, with the

exoskeleton (for example, Flyjecket [10]), wearable glove [11], EMG sensor [5], and wearable watch (such as the EMPA watch) having been reported for application to human–UAV interaction (as shown in Figure 1). Although such methods are intuitive for the user, they usually require expensive devices and lack the ability to differentiate between an unconscious action and a similar pre-defined action.

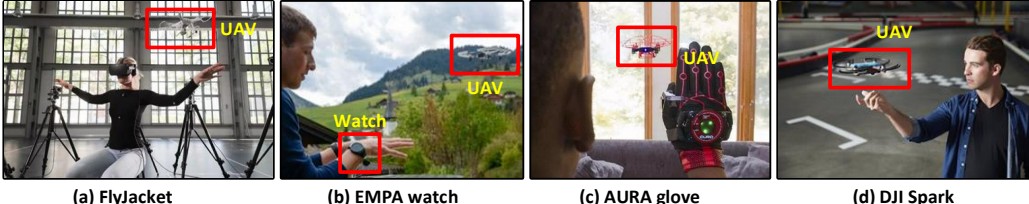

| (a) FlyJacket | (b) EMPA watch | (c) AURA glove | (d) DJI Spark |

**Figure 1.** Recently developed system for controlling a UAV in different ways.

Compared to the traditional remote controller, recently developed user-friendly remote controllers (such as those depicted in Figure 2a,b) can aid a human beginner to fly a UAV with little training. The UAV pilot can easily control the UAV and complete the path planning using the touch-screen device. The smartphone-based controlling method may be considered as a variation of the traditional remote controller; however, with this method, the distance between the UAV and its pilot is usually less than 100 m, owing to the limitations of WiFi signal transmission [12]. In general, such methods are the most popular solutions and have been accepted by the community. However, in the case of a UAV rescue operation, as illustrated in Figure 2c, since the external pilot is usually far away from the person being rescued (hundreds of meters or even miles away), it will become time consuming and difficult for the operator to finish the rescue operation by using the remote controller for lacking some important information such as: wind direction and speed of rescue site and the 3D position relationship between the person to be rescued and the UAV platform. However, the person being rescued knows these information better than the pilot and the direct interaction between him and the UAV is considered as a better way to rescue person. On such consideration, the human–UAV interaction is far beyond flying the UAV. In such case, as the person needing rescue cannot access a remote controller, the UAV needs to understand the human action (or gesture) by itself and not through the UAV pilot (particularly in the case of tele-operation, where the time delay between the UAV and its pilot is critical).

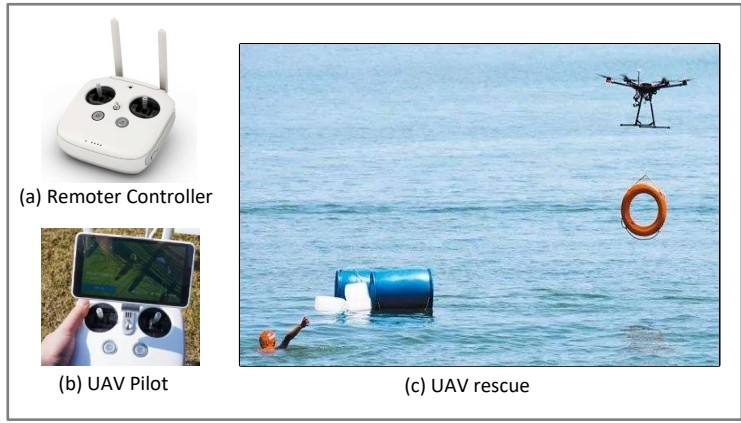

**Figure 2.** Human-UAV interaction is more than flying a UAV (like (**a**,**b**). In the case of UAV rescue (**c**), the UAV needs to understand the action of person who needs to be rescued).

Compared with the remote controller, the gesture/action-recognition-based human-UAV interaction is the more natural and intuitive method that *"....could improve flight efficiency, reduce errors, and allow users to shift their attention from the task of control to the evaluation of the information provided by*

*the drone."* Ref. [10] Owing to the rapid progress in computer vision and machine learning (mainly deep learning) algorithms in the past decade, vision-based gesture and action recognition [13,14] has undergone significant improvements. By inputting massive training samples into a three-dimensional (3D) convolutional neural network (CNN), in [15], the authors achieved impressive experimental results in a gesture recognition task under a real scenario. Further improvements in action recognition using the long short-term memory (LSTM) spatio-temporal network [16], two-stream network [17], and recurrent neural network (RNN) [18] have also been reported. Recent research demonstrated that the graph convolutional network (GCN) method [19] is superior to other action recognition methods by applying the space–time graph to human skeleton nodes to determine the accurate action category.

Benefitting from the recent developments in computer vision technology, the DJI spark drone has been reported as the first commercial UAV that can understand human gestures with four functions: snapshot, landing/departing, tracking the human hand, and going away. As the maximum working range of tracking the human hand (and going away) is usually less than 3 m, this system is limited to mini drones, and is not suitable for larger industrial UAVs. Furthermore, its recognition rate for snapshot gestures is approximately 60~90% (according to our test results with a beginner), which is far from being suitable for applications in controlling UAV movement. Other researchers have attempted to apply recent deep learning-based gesture/action algorithms (such as LeapMotion [2,20], OpenPose+LSTM [21,22], YOLO [23] and P-CNN [24]) to human–UAV interaction. The general recognition accuracy of these works varied from 90% to 92%, which is not sufficient for controlling a UAV. Furthermore, it has always been assumed that only one person will be captured by the camera.

Using a gesture recognition or action detection method can hardly satisfy the crucial UAV-controlling requirements (the recognition accuracy should be over 99%) owing to the limited discriminant ability [25]. Therefore, in this study, we developed a deep learning-based gesture–action joint detection algorithm for human–UAV interaction. The deep learning-based action detection and multi-feature based cascade gesture recognition algorithms were combined by a cross-validation process to obtain the corresponding operation command. As the flight strategy of a UAV will be changed according to different working conditions, we used the learning-based scene classification algorithm and recognized operation command to produce a unique flight command. Such a process can enable the UAV understand the command from the UAV pilot and automatically adjust its flight strategy according to the flight conditions, simultaneously.

## 2. Proposed System

### 2.1. Overview of Proposed System

**Motivation:** The interactions between humans and robots have become increasingly important. Therefore, in many cases, an intelligent robot needs to understand the commands or wills of humans more intuitively. Figure 3 presents examples in which human–robot interaction is quite an important ability for the robot. Moreover, such interaction should not be unidirectional communication, but rather bidirectional. This means that robots should not only be remote controlled by humans, but also understand the human will and take the corresponding actions autonomously.

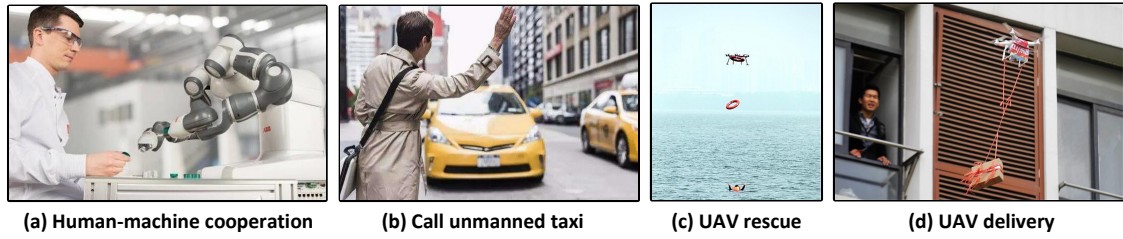

**(a) Human-machine cooperation**　　**(b) Call unmanned taxi**　　**(c) UAV rescue**　　**(d) UAV delivery**

**Figure 3.** Gesture/Action recognition is the intuitive way for human-robot interaction under many tasks.

We proposed a gesture–action joint recognition method for human–robot interaction and tested it on a real UAV platform. Such an ability is also important for a UAV to complete numerous difficult

tasks. For example, in Figure 3c,d, the person needs to be rescued or the package receiver does not contain a remote controller, and the gesture/action-based human–UAV interaction will play an important role in the rescue operation (or delivery task). Furthermore, we believe that such an interaction method is not limited to UAVs, but offers potential applications for all types of robots (such as industrial robots and unmanned taxis, as illustrated in Figure 3a,b).

**Problem statement of current methods:** As illustrated in Figure 4, even for state-of-the-art methods, neither spatio-temporal action recognition nor gesture recognition algorithms are sufficient to be directly applied to human–robot interaction [26]. This is because the action recognition algorithm may suffer from: (1) its limited classification ability (up to 95 % recall rate, as described in [19]); and (2) a similar action caused by another person (as in Figure 4a). Moreover, the gesture recognition algorithm may incorrectly produce a false alarm in the background (Figure 4b is the result of *YOLO V3* [27]). Therefore, in this paper, we combined the deep learning based action recognition with a multi-feature cascade gesture recognition algorithm for human-UAV interaction, because the competition of two algorithms may greatly reduce the false alarm of whole system.

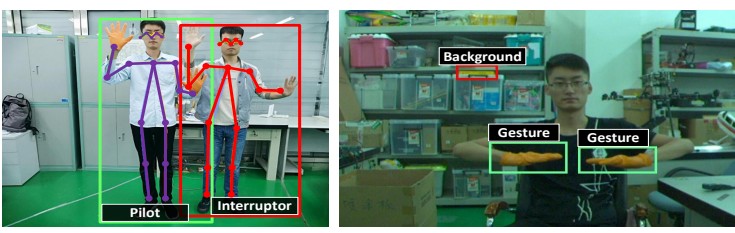

( a ) Pose estimation result of Alpha-Pose    ( b ) Gesture recognition result of YOLO V3

**Figure 4.** The conventional action or gesture recognition algorithms are difficult to be directly applied for human-robot interaction. (**a**) Action recognition result of Alpha-Pose [28]; (**b**) Gesture recognition result of *YOLO V3* [27].

The main contributions of this work include the following: (1) a scene-understanding algorithm is applied to enable the UAV to change its flight strategy automatically, according to its working conditions; and (2) a cross-validation process is applied to combine the results of the spatio-temporal action recognition and static gesture recognition, which can significantly reduce the false alarms of the visual recognition algorithms. Compared with the present state-of-art gesture/action-recognition-based human-UAV interaction systems, the proposed method is superior to them in three ways as: (1) intelligent ability to adjust the flight strategy according to the flight condition; (2) increased correct recognition accuracy up to over 99% (the others vary between 90∼92%); (3) working stably when multiple persons doing similar action are captured in the image (the others just assume one person is captured).

As illustrated in Figure 5, the proposed system is composed of five modules: a scene-understanding module, pilot detection module, action detection and recognition module, gesture recognition module, and joint reason and control module. After the operator is found through the pilot detection module, his or her action will be detected and recognized by a deep learning-based action classification process. Meanwhile, in order to reduce the false alarms produced by the action recognition, a multi-feature cascade gesture recognition method is applied, based on the fact that both the action recognition and gesture recognition producing false alarms is a low-probability event. The combination of each gesture and action corresponds to a unique flight action. As the UAV generally uses different flight strategies under various working conditions, we introduce the deep learning-based scene-understanding algorithm into the flight-controlling module with the combination of the flight action produced by the visual action and gesture recognition.

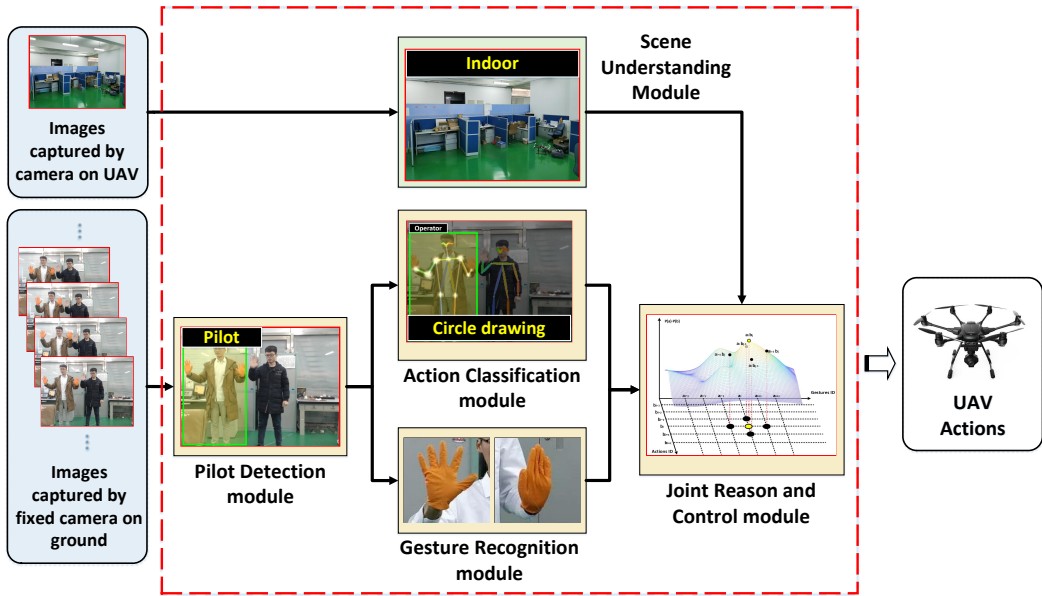

**Figure 5.** Structure diagram of the system.

## 2.2. Scene Understanding for Uav Control

**Motivation:** Figure 6 presents examples that effectively demonstrate why scene understanding is an important ability for a UAV to contain to follow the different working conditions. For example, when landing or delivering baggage on a ship, even if the same flight action is used, the flight strategy selected by a UAV should be completely different from that it applies when working at the polar station (a detailed definition of the parameters in the flight strategy can be found in Table 1). Based on this observation, we introduce the deep learning-based scene-understanding algorithm into the UAV control system.

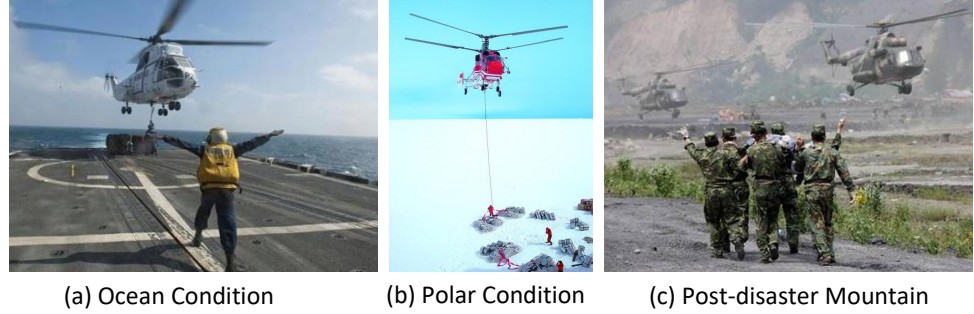

(a) Ocean Condition      (b) Polar Condition      (c) Post-disaster Mountain

**Figure 6.** Under different working conditions, a UAV should choose different flight strategy.

We selected the well-known Resnet-101 [29] to achieve the scene-understanding process, owing to its outstanding classification results with a much more complex and deeper network structure (as illustrated in Figure 7). The Resnet-based scene classifier was trained with the Places365 dataset (its implementation is presented in the following section). Further information on Resnet is beyond the scope of this paper, and its detailed description can be found in [29]. In our system, as illustrated in Figure 8, the scene classifier was applied to the aerial images captured from the UAV every 3 s. To reduce the effects of random false alarms, a statistics voting process was applied: from the five successive classification results, the most frequently appearing result was considered as the final scene-understanding result. The corresponding flight strategy for the scene-understanding result of our UAV control system is presented in Table 1.

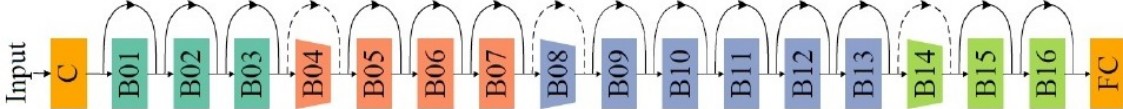

**Figure 7.** The structure of Resnet101 [29].

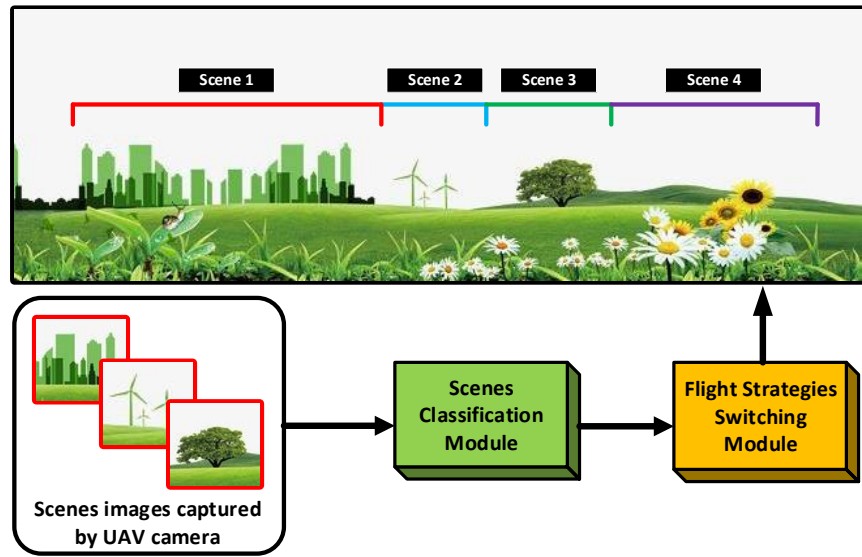

**Figure 8.** Diagram of scene understanding to automatically adjust the flight strategy under different working conditions.

**Table 1.** UAV flight strategy corresponds to scene understanding result in our research.

| Scene Label \ Strategy | Top Speed (m/s) | Acceleration (m/s$^2$) | Maximum Height (m) | Possible to Land |
|---|---|---|---|---|
| indoor | 1 m/s | 0.25 | 2 | Yes |
| buildings | 2 m/s | 0.50 | 10 | Yes |
| woods | 3 m/s | 1.00 | 20 | Yes |
| factory | 2 m/s | 0.75 | 20 | Yes |
| square | 4 m/s | 1.00 | 50 | Yes |
| water surface | 5 m/s | 1.50 | 50 | No |

### 2.3. Pilot Detection Module

As indicated in Figure 4, action or gesture recognition algorithms are not suitable for direct application to control the UAV, because they may be confused by a similar action caused by another person or produce false alarms on background objects. To address these problems, we developed the pilot detection module to: (1) isolate the real UAV pilot from another person; (2) extract the skeleton points from the real pilot; and (3) determine the correct hand regions for gesture recognition.

Figure 9 presents an overall diagram of the pilot detection process. Firstly, people in the images will be extracted by YOLOv3 [27], where multiple persons may be detected from the image sequence. Thereafter, to maintain the correct correspondence of multiple people, a deep sort tracking algorithm [30] is used to provide the correct ID of each detected person continuously. The AlphaPose [28] pose estimation algorithm, which is based on Faster R-CNN [31], is applied to the regions of detected people to extract the key skeleton points of human bodies, through which the hand areas can easily be located. Finally, the UAV pilot will be separated from the other persons by

applying a color filter to their hand regions, because the real pilot is requested to wear orange-colored gloves. The extracted key skeleton points of the pilot are used for the action recognition detailed in the following section.

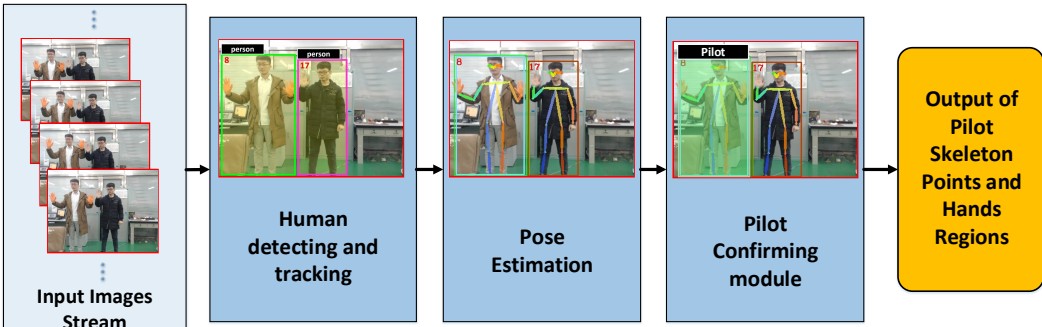

**Figure 9.** Diagram of pilot action detection module.

Figure 10 presents an example of our pilot detection module, where the background contains similar objects such as human hands, and another human performs similar actions to the real UAV pilot. By using the pose estimation and color similarity measurements, our system can successively remove the noisy background components and distinguish the real pilot from the other persons.

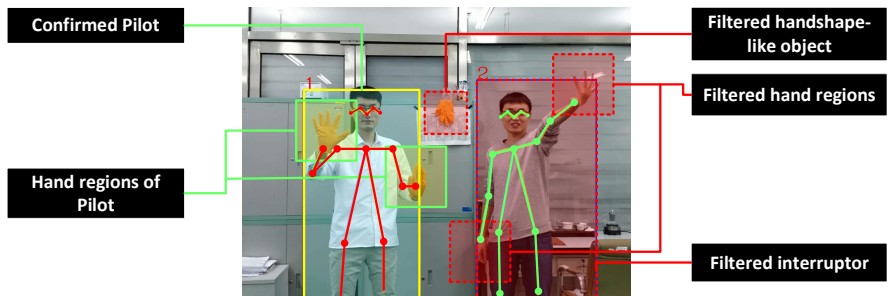

**Figure 10.** The real UAV pilot and his hands will be correctly extracted and the other human and background noise will be ignored.

*2.4. Action Recognition Module*

Compared to traditional methods that use spatio-temporal features to describe human motion (such as HOGHOF [32], STHOG [33,34], STGGP [35] and 3DHOG [36]), recently developed GCNs can recognize human actions more accurately by using the key points extracted from their bodies as the input features. As illustrated in Figure 11, after obtaining the key skeleton points from a real UAV pilot, all of these sequential key points will be transmitted to the spatial–temporal GCN (ST–GCN [19]) module for the action recognition task.

$$f_{out}^t(\mathbf{x}_c) = \sum_{h=1}^{K} \sum_{w=1}^{K} f_{in}(\mathbf{S}(\mathbf{x}_c), h, w) \cdot \mathbf{W}(h, w) \tag{1}$$

Here, $\mathbf{S}$ represent the sampling function used to enumerate the neighbors of location $\mathbf{x}$, and $\mathbf{W}$ represent the weight function which provides a weight vector for computing the inner product with the sampled input feature vectors within the convolutional kernel. $h$ and $w$ represent the pixels within the kernel. In this paper, six kinds of actions (named as $a_i, i = 1 \sim 6$) are used to control the flight

action of UAV. The probability of an input action $a^{(in)}$ to be classified as action $a_i$ is computed from a softmax function $P(a_i|a^{(in)})$ through $M$ successive frames as:

$$P(a_i|a^{(in)}) = softmax(\sum_{t=1}^{M} \sum_{c=1}^{N} f_{out}^t(\mathbf{x}_c)). \tag{2}$$

In the case that the maximum output value of this function is over a predefined threshold, the input action will be considered as the corresponding action. A more detailed explanation of the ST–GCN can be found in [19].

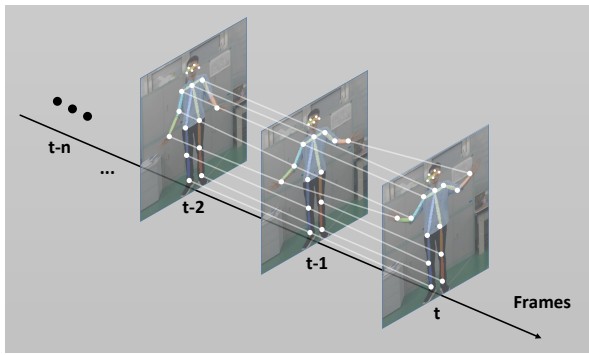

**Figure 11.** Illustration of the ST-GCN [19], where the spatial temporal graph of a skeleton sequence are used to classify the action of a human.

### 2.5. Gesture Recognition Module

As illustrated in Figure 4b, even if training takes place with massive samples, false alarms may still arise owing to the complex background conditions or the training samples not being able to cover all possible variations in human skin color and shape, among others. Moreover, different people may perform the same gesture in varying manners as a result of their unique habits.

Based on these considerations, instead of using offline training-based algorithms, we propose an online visual gesture recognition method, in which the personal features of each pilot will be extracted and sorted in his personal feature library before the entire system is operating. The overall processing diagram of this module is presented in Figure 12.

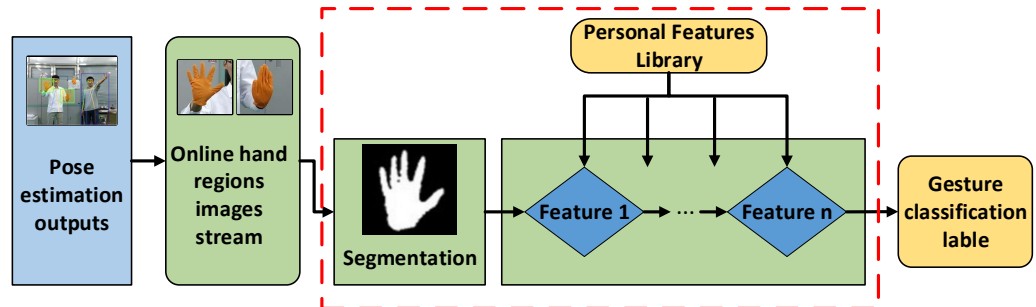

**Figure 12.** Gesture recognition module.

Within the region of interest (ROI) extracted by the operation detection module, the hand region will be segmented according to the color similarity with the adjustable threshold to follow the variation of illumination and distance. Three kinds of features (expressed as $\mathbf{x} = [x_i, i = 1\sim3]$) will be extracted from the ROI as illustrated in Figure 13, where the convex hull and dig angle usually change greatly due to the habit of different person and these feature can not only distinguish hands from other background objects but also from person to person. In this research, six kinds of gestures are applied

for controlling the UAV and each one is expressed as ($g_j$, j = 1~6). Therefore, the features of each gesture could be defined as ($x_i^j$, i = 1~3, j = 1~6). In the personal feature library, for each pilot, the mean and standard variation of each gesture feature will be expressed as ($\mu_i^j(reg)$, $\sigma_i^j(reg)$)

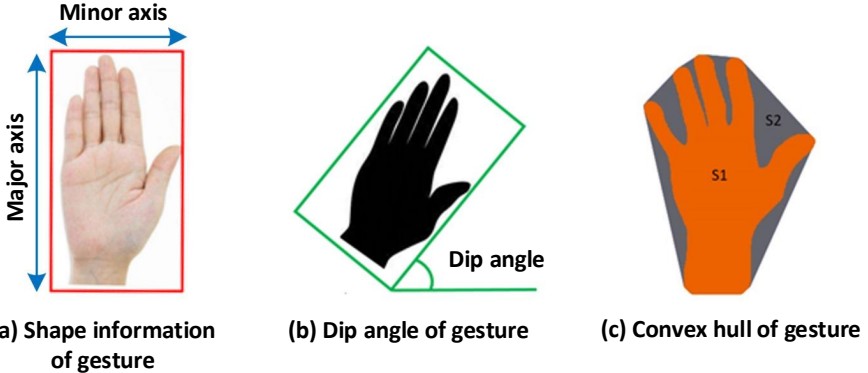

**(a) Shape information of gesture**      **(b) Dip angle of gesture**      **(c) Convex hull of gesture**

**Figure 13.** Description for features.

As for the features ($x_i^{(in)}$, i = 1~3) of an input gesture ($g^{(in)}$) which is segmented from the hand ROI, a multi-feature cascade classification process is applied to identify its gesture category by the following equations as:

$$Gesture(ID) = \arg\max_{1 < j < 6} \left( P(g_j | g^{(in)}) \right), \tag{3}$$

$$P(g_j | g^{(in)}) = P(x_1^j | x_1^{(in)}) * P(x_2^j | x_2^{(in)}) * P(x_3^j | x_3^{(in)})), \tag{4}$$

$$P(x_i^j | x_i^{(in)}) = exp^{\frac{-(x_i^{(in)} - \mu_i^j(reg))^2}{\sigma_i^j(reg)^2}}. \tag{5}$$

Here, $P(x_i^j | x_i^{(in)})$ denotes the similarity between $i_{th}$ feature of an input gesture $g^{(in)}$ and that of gesture $g_j$ registered in the personal feature library. $P(g_j | g^{(in)})$ represents the similarity from input gesture $g^{(in)}$ to the predefined gesture $g_j$. The category of $g^{(in)}$ is obtained by maximizing Equation (3) and such maximum similarity is also beyond a predefined threshold. According to Equation (4), a single low feature similarity between input gesture $g^{(in)}$ and $g_j$ will lead to its rejection to gesture $g_j$, such process could be considered as the well-known cascade classification procedure. Detailed description about the color segmentation and the computation for ($\mu_i^j(reg)$, $\sigma_i^j(reg)$) could be found in [37,38].

### 2.6. Joint Reason and Control Module

As shown in Figure 14, some gestures and actions are quite similar to each other and they are difficult to be identified by either the offline learning based or the online gesture recognitions. In Figure 14a, the two gestures are quite similar and in (b) the actions of hands up and that of drawing circle are almost the same. In such cases, the performance of gesture and action recognition algorithms needs to be improved so as to be applicable for controlling the UAV.

In order to solve this problem, we introduced a joint cross validation over the action and gesture recognition results. In this research, we separately selected six kinds of actions and gestures to control the flight action of a UAV. As shown in Figure 15, all the gestures and actions are arranged into a circle ring where each gesture is expressed as $g_j$ (j = 1~6) and each action is denoted as $a_i$ (i = 1~6). In order to separate similar gestures or actions from each other, each node is arranged by the following rules as: as for an input gesture $g^{(in)}$ and action $a^{(in)}$ with their correct category as $g_j$ and $a_i$, their probability to the other nodes should be $P(g_j | g^{(in)}) \gg P(g_{j\pm1} | g^{(in)}) \gg P(g_{j\pm2} | g^{(in)})$...

and $P(a_i|a^{(in)}) \gg P(a_{i+1}|a^{(in)}) \gg P(a_{i+2}|a^{(in)})$. When $P(g_j|g^{(in)}) \sim P(g_{j+1}|g^{(in)})$ or $P(a_i|a^{(in)}) \sim P(a_{i+1}|a^{(in)})$, it means the action-gesture recognition results are un-reliable and we will send a special command to the UAV to keep it hovering.

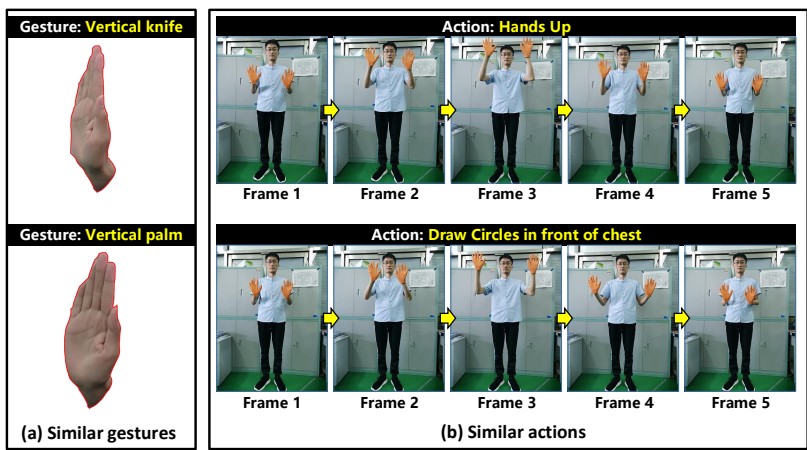

**Figure 14.** There are some gestures and actions quite similar to each other. In (**a**), two gestures are quite similar, and in (**b**), in five successive images, the human action in four images looks almost the same as others.

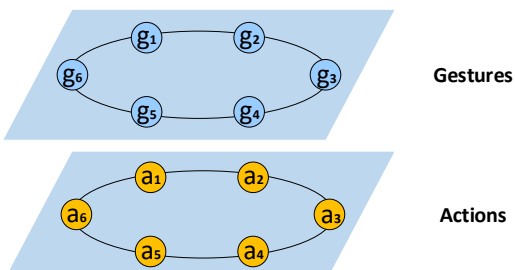

**Figure 15.** Actions and gestures confusion ring sequence: $g_1$: Rock, $g_2$: Vertical palm, $g_3$: Vertical knife, $g_4$: Paper, $g_5$: Horizontal palm, $g_6$: Horizontal knife. Description for actions: $a_1$: Waving left hand, $a_2$: Hands up, $a_3$: Draw circles in front of chest, $a_4$: Swing hand up and down, $a_5$: Waving right hand, $a_6$: Draw circles over head.

Such arrangement could guarantee that, as for an input action (or gesture) $a^{(in)}$ (or $g^{(in)}$), even if the classification for it is wrong, the obtained action/gesture category ($i \pm 1$ or $j \pm 1$) could still be the nearest neighbor to the correct one.

Then, as shown in Figure 16, the sparse coding for the combination of action and gesture has been set up, where the eighteen orange blocks mean the correct codebook that the UAV pilot will use to control the UAV and the other blocks will not be applied in our system. As for the input action $a^{(in)}$ and gesture $g^{(in)}$, let $id_a$ represent their classification result that can maximize Equation (2) and $id_g$ be the output from Equation (3). The corresponding combination of $g_{id_g}a_{id_a}$ could be checked as follows:

$$g_m a_n = \begin{cases} g_{id_g}a_{id_a} & if \quad g_{id_g}a_{id_a} \in codebook \\ \arg\max\left(P(g_{id_g \pm 1})P(a_{id_a \pm 1})\right) & else \end{cases} \tag{6}$$

(if $id_{g/a} + 1 > 6$, set $id_{g/a} + 1 = 1$; if $id_{g/a} - 1 = 0$, set $id_{g/a} - 1 = 6$.)

And $g_m a_n$ represents their final ID in the codebook through cross validation process. In the case that $g_{id_g}a_{id_a}$ lies in the orange block of Figure 16, they will be considered as the correct result and translated to the corresponding flight action in Table 2. However, when $g_{id_g}a_{id_a}$ is out of the codebook

(means they lie in the white blocks), it means either the gesture or the action recognition result is incorrect. The correction for such result will performed by search for its four-neighbor block and find the very one that can maximum $(P(g_{id_g \pm 1})P(a_{id_a \pm 1}))$.

**Figure 16.** Sparse coding of Gestures-Action combinations. (The orange blocks are selected into our codebook).

**Table 2.** UAV flight action obtained through the cross validation process.

| Hybrid Coding | UAV Corresponding Action | Hybrid Coding | UAV Corresponding Action |
| --- | --- | --- | --- |
| $g_1 a_1$ | Fly towards left | $g_2 a_4$ | Fly towards right |
| $g_3 a_1$ | Rotate left | $g_4 a_4$ | Rotate right |
| $g_5 a_1$ | Fly upward | $g_6 a_4$ | Fly down |
| $g_2 a_2$ | Fly foreword | $g_1 a_5$ | Fly backword |
| $g_4 a_2$ | Hover | $g_3 a_5$ | Circling |
| $g_6 a_2$ | Draw square | $g_5 a_5$ | Draw S |
| $g_1 a_3$ | Speed up | $g_2 a_6$ | Speed down |
| $g_3 a_3$ | (Undefined) | $g_4 a_6$ | (Undefined) |
| $g_5 a_1$ | (Undefined) | $g_6 a_6$ | (Undefined) |

Figure 17 show an illustration of how this cross validation could correct the wrong coding. Assuming that the gesture recognition result $g_{id_g}$ is correct and the action recognition result $a_{id_a}$ is incorrect, where the correct one should be $a_{id_a-1}$. Through Equation (6), our validation will compare its four coded neighbors as $(P(g_{id_g})P(a_{id_a-1}), P(g_{id_g})P(a_{id_a+1}), P(g_{id_g-1})P(a_{id_a}), P(g_{id_g+1})P(a_{id_a}))$. Since the input gesture is correctly classified which means $P(g_{id_g}|g^{(in)}) \gg P(g_{id_g \pm 1}|g^{(in)})$, it will be easy for our system to reject $P(g_{id_g-1})P(a_{id_a}), P(g_{id_g+1})P(a_{id_a})$. Because the true category of action is $a_{id_a-1}$, the probability $P(g_{id_g})P(a_{id_a+1})$ will be quite low and rejected. In this way, the maximum probability should be $(P(g_{id_g})P(a_{id_a-1})$ and the true action ID is $a_{id_a-1}$.

The corresponding relationship between the action–gesture combination and UAV flight action is displayed in Table 2 (the four undefined combinations are for pilot customization). The final flight command will be the combination of the flight strategy obtained from the scene understanding and flight action resulting from the cross-validation over the action–gesture recognition results.

After receiving the flight command sent from the PC workstation, the H-infinite [39,40] algorithm is applied to control the flying UAV. The H-infinite technique is superior to classical control techniques in its ability to solve many problems involving multivariate systems with cross-coupling among channels.

As the scene-understanding algorithm is applied in our system and the flight strategy will change from scene to scene, it should be noted that, even if the UAV pilot sends the same commands to the UAV, the UAV may perform its final flight action in different manners according to its flying environment.

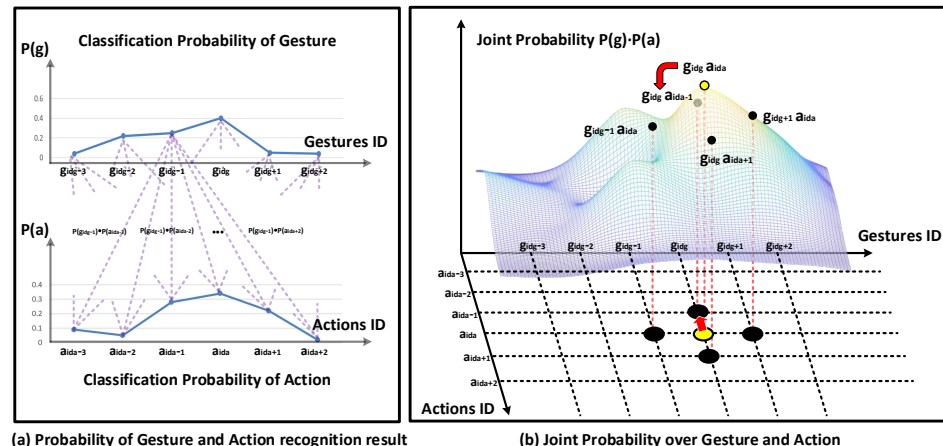

**Figure 17.** Illustration of the cross validation process. (**a**) shows the probability distribution of action and gesture recognition results. (**b**) shows how our cross validation process could correct the erros in (**a**). Here, the yellow point represent the wrong coding combination when either action or gesture recognition results is wrong. Block points refer to the correct coding that should be used to control UAV.

*2.7. Safety Measurements*

As the success of our system is based on the assumption that either the action or gesture recognition result is correct, in the case that neither is correct, we select the following four measurements to minimize the flight risk: (1) virtual fence (also called electronic fence as illustrated in Figure 18) to restrict the flight area; (2) a limitation of the UAV top speed, acceleration, and flight height (as per Table 1); (3) automatic hovering when all data are lost; and (4) the pilot continuing to watch the screen of the ground station and switch to the manual remote-controlling model when the UAV appears to lose control.

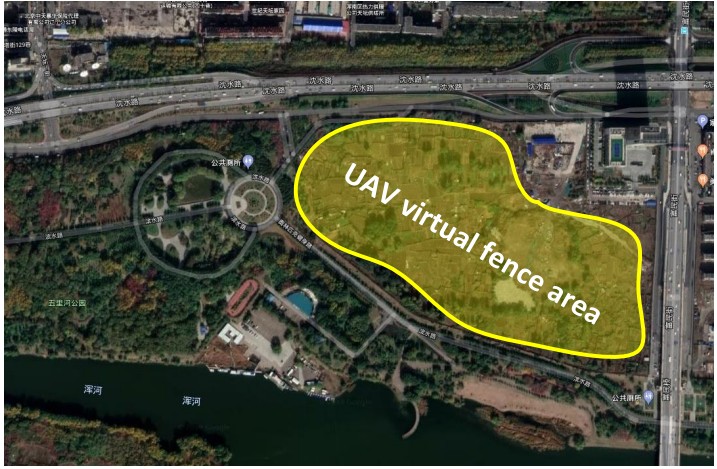

**Figure 18.** Virtual fence to restrict the flight area.

## 3. Experiments

To evaluate the performance of our algorithm, we performed four experiments to test each part of the system: (1) a comparative experiment for the action recognition module; (2) a comparative experiment for the gesture recognition; (3) a scene-understanding experiment; and (4) a real flight experiment with the UAV. All four experiments were performed on a PC with an Intel Core i7 3.40GHz CPU, 64 GB memory, and GPU GTX1080ti. The deep learning algorithms were compiled with CUDA 10.1 under the Pytorch 0.4.1 environment. The hardware used in these experiments is illustrated in

Figure 19. The test UAV was a self-assembly six-rotor UAV with a wheelbase of 55 cm, height of 40 cm, and weight of 1.2 kg. The camera in this system was a Basler industrial camera with a resolution of 640 × 480 pixels.

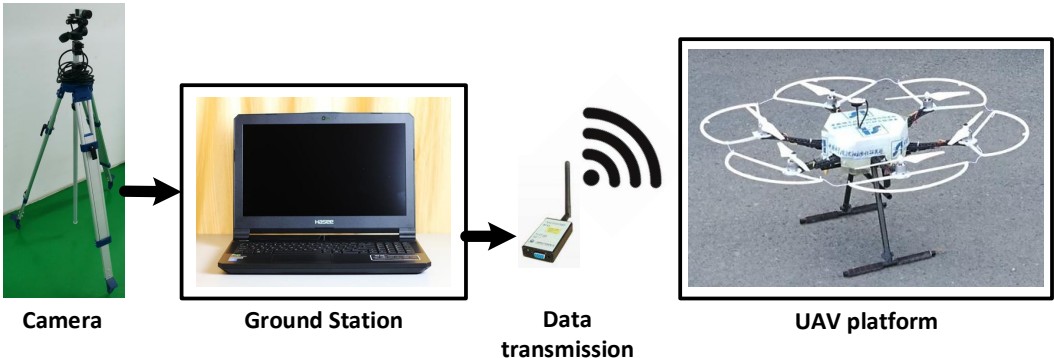

**Figure 19.** Hardware of our system.

In the following sections, besides the public dataset, we also prepared our own dataset for action and gesture recognition experiments, where all the data were collected by 3 inviduals and categorized into 6 classes for action recognition and 6 classes for gesture recognition. All the classes were created according to the action/gesture in the images but not the collectors. 12 users were requested to perform the following simulation experiment and 5 of them were selected to perform the final flight experiment on the real UAV platform.

### 3.1. Comparative Experiments for Action Recognition

To evaluate the performance of the action recognition algorithm, we compared four types of action recognition algorithms: ST–LSTM+TS [41], temporal convolutional networks [42], P-CNN [24] and the ST–GCN [19] applied to our system. The test datasets were composed of two parts: the well-known NTU dataset [43] and our own dataset. The comparative results are presented in Tables 3 and 4.

**Table 3.** Comparative experiments on the NTU dataset (57 k training samples and 5 k test samples).

| Algorithm | Accuracy Rate |
|---|---|
| ST-LSTM + TS [41] | 69.2% |
| Temporal Conv [42] | 74.3% |
| ST-GCN [19] | **82.5**% |

**Table 4.** Comparative experiments on our 6-class action dataset (5.2 k training samples and 600 test samples).

| Algorithm | Accuracy Rate |
|---|---|
| AlphaPose [28] + ST-LSTM + TS [41] | 89.2% |
| AlphaPose [28] + Temporal Conv [42] | 93.5% |
| AlphaPose [28] + ST-GCN [19] | **95.8**% |
| OpenPose [21] + ST-LSTM + TS [41] | 88.8% |
| OpenPose [21] + Temporal Conv [42] | 94.2% |
| OpenPose [21] + ST-GCN [19] | 95.5% |
| P-CNN [24] | 79.5% |

In the NTU dataset, all of the algorithms were trained using 57,000 samples and tested with 5000 samples (with 40 users collected from 79.5 h-long video). In our dataset, there were 5200 training samples and 600 test samples (with 3 users collected from 85-min video) with six types of actions, and all of the test action recognition algorithms were combined with OpenPose [21] and Alpha-Pose [28], which can provide skeleton points for further recognition. The ST–GCN algorithm was superior to all of the other algorithms.

### 3.2. Comparative Experiments among Gesture Recognition Algorithms

To confirm the effectiveness of our gesture recognition algorithm, we compared our method with five other gesture recognition algorithms, namely methods based on: (1) the FPN [44]; (2) RefineDet [45]; (3) Faster R-CNN [31]; (4) convolutional pose machines [46] and Resnet101 [29]; and (5) YOLOv3 [27]. We collected four image groups under different conditions: i) simple indoor background; ii) complex indoor background; iii) simple outdoor background; and iv) complex outdoor background. We prepared two dataset for evaluation as: NYU hand pose dataset [47] containing 52,360 training samples and 22,440 test samples from 11 users; our own dataset containng 5272 training samples (2000 positive and 3272 negative) and 870 test samples from 12 users. All of the methods were trained and tested with the same samples.

To describe the performance of all compared methods, the recall precision curve (RPC) was applied for the evaluation, in which the recall rate and precision are calculated as follows:

$$RecallRate = \frac{\sum(CR)}{\sum(GT)} \tag{7}$$

$$Precision = \frac{\sum(CR)}{\sum(TRR)} \tag{8}$$

Here, "$\sum(CR)$" means the number of correct recognition results; "$\sum(GT)$" refers to the number of ground truth, and "$\sum(TRR)$" corresponds to the total number of recognition results.

The RPC analysis results of the different methods are presented in Figures 20 and 21, and Tables 5 and 6. The performance of our gesture recognition algorithm was superior to those of the compared methods using our dataset, and quite similar to those of RefineDet [45] and YOLOv3 [27] using the NYU dataset. The main reason for the difference between the results of our dataset and the public dataset is that the background of our dataset was more complicated than that of the public dataset, in which hand-like objects in the background could easily be recognized as hands.

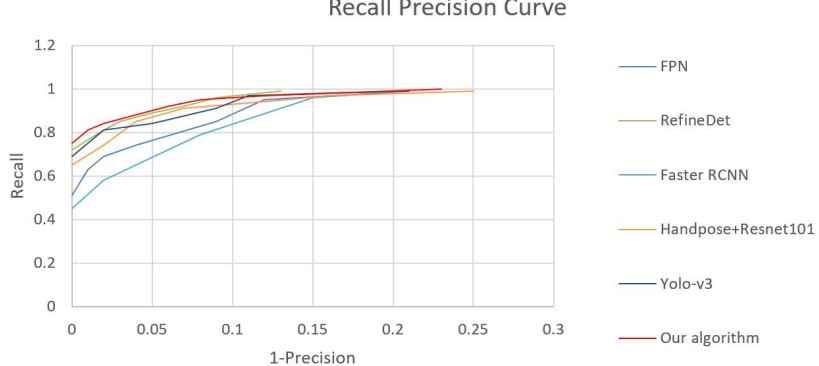

**Figure 20.** The RPC analysis results based on NYU hand gestures dataset [47].

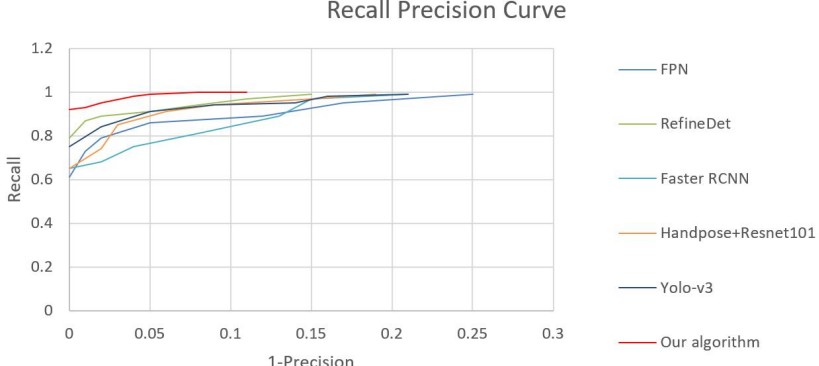

**Figure 21.** The RPC analysis results based on our own dataset.

**Table 5.** Recall Rate of different algorithm when on the NYU test data when precision is set as 0.98. ("P" represents gesture Paper, "HP" represents gesture Horizontal Palm, "HK" represents gesture Horizontal Knife, "VP" represents gesture Vertical Palm, "VK" represents gesture Vertical Knife, "R" represents gesture Rock).

| Algorithm / Gestures | P | HP | HK | VP | VK | R |
|---|---|---|---|---|---|---|
| FPN [44] | 0.71 | 0.65 | 0.62 | 0.66 | 0.70 | 0.75 |
| RefineDet [45] | 0.78 | **0.88** | 0.75 | 0.78 | 0.79 | 0.74 |
| Faster RCNN [31] | 0.62 | 0.58 | 0.65 | 0.55 | 0.65 | 0.63 |
| Handpose [46] + Resnet101 [29] | 0.75 | 0.84 | 0.67 | 0.72 | 0.69 | 0.66 |
| Yolo-v3 [27] | 0.84 | 0.86 | **0.89** | 0.82 | 0.85 | 0.80 |
| Our algorithm | **0.85** | 0.83 | 0.88 | **0.92** | **0.86** | **0.88** |

**Table 6.** Recall Rate of different algorithm when Precision is 0.98.(trained and tested by our datasets).

| Algorithm / Gestures | P | HP | HK | VP | VK | R |
|---|---|---|---|---|---|---|
| FPN [44] | 0.78 | 0.77 | 0.80 | 0.74 | 0.82 | 0.77 |
| RefineDet [45] | 0.78 | 0.81 | 0.79 | 0.83 | 0.79 | 0.85 |
| Faster RCNN [31] | 0.60 | 0.61 | 0.58 | 0.69 | 0.65 | 0.57 |
| Handpose [46] + Resnet101 [29] | 0.71 | 0.75 | 0.68 | 0.69 | 0.79 | 0.71 |
| Yolo-v3 [27] | 0.81 | 0.85 | 0.76 | 0.83 | 0.82 | 0.86 |
| Our algorithm | **0.97** | **0.97** | **0.94** | **0.98** | **0.95** | **0.96** |

### 3.3. Experiments of Scene Understanding Module

To evaluate the performance of the scene-understanding algorithms, comparative experiments were performed among ResNet-101 [29], AlexNet [48], Inception-v3 [49], and VGG-19 [50]. All of the algorithms were tested using the public Places365 dataset [51], which covers 365 scene categories with 10 million images. All of the algorithms were trained with 9 million samples from the Places365 training dataset and tested with 1 million samples. Table 7 presents the comparative experimental results among the four scene-understanding algorithms, where VGG-19 was superior in Top-1 and ResNet-101 was the best classifier in Top-5. According to the flight strategy of Table 1, we further categorized the 365 classes into 6 classes, the experimental results of which are displayed in Table 8.

According to Table 8, ResNet-101 outperformed the other scene-understanding algorithms when the 365 image classes were further clustered into six classes as per the flight strategy described in Table 1. Meanwhile, we also established our own test dataset by collecting 160 real aerial images covering six

types of scenes, namely indoor, factory, square land, water surface, and buildings. Figure 22 illustrates several scene classification results of ResNet-101 over the real aerial images captured from our UAV. In our dataset, the classification accuracy of ResNet-101 was 96.25%, which was quite close to its performance in the public dataset.

**Table 7.** Comparative experimental results of scene understanding algorithms on datasetPlace365. (365 classes).

| CNN Model | Accuracy Top-1 | Accuracy Top-5 |
|---|---|---|
| AlexNet [48] | 53.17% | 82.89% |
| Inception-v3 [49] | 53.63% | 83.88% |
| VGG-19 [50] | **55.24**% | 84.91% |
| ResNet-101 [29] | 54.74% | **85.08**% |

**Table 8.** Comparative experimental results of scene understanding algorithms when dataset Place365 is categorized into 6 classes according to Table 1.

| CNN Model | Accuracy Top-1 |
|---|---|
| AlexNet [48] | 93.76% |
| Inception-v3 [49] | 95.28% |
| VGG-19 [50] | 96.41% |
| ResNet-101 [29] | **97.02**% |

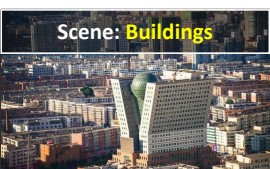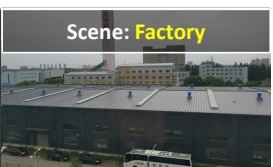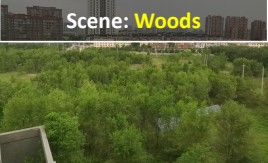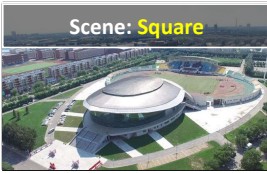

**Figure 22.** Scenes understanding results of ResNet-101 [29] over real aerial images.

*3.4. Experiments of Action–Gesture Joint Uav Control*

The experiments of the overall proposed system were composed of two parts: a (1) simulation experiment of the joint cross-validation over the action–gesture recognition; and (2) a real flight experiment, performed by applying the proposed system to control a UAV under a real urban scene. Figure 23 presents an example of the simulation experiment of our system, with a test video of 2.5 h in length (including a total of approximately 270,000 images with 6 types of actions and gestures). The UAV pilot sent the flight command through gestures and actions, and only the correctly recognized gestures and actions could lead to the correct flight command.

It can be observed from Table 9 that that, when combining the actions and gestures directly, the accuracy of the combination result was lower than the correct rates of the actions and gestures, respectively. This is because such a combination could be considered as the inner product of the action and gesture recognition results, and the false alarms from either result would lead to incorrect judgment. However, after applying the proposed joint cross-validation over the action–gesture recognition results, the final recognition accuracy achieved was 99.4% (tested in our offline dataset).

**Table 9.** The recognition accuracy of each module in our system when Recall Rate is 0.98.(trained and tested by our datasets, *accuracy* just refer to the recalll rate of Equation (7)).

| Action Accuracy | Gesture Accuracy | Action-Gesture Combination | Action-Gesture Cross-Validation |
|---|---|---|---|
| 95.8% | 96.7% | 92.5% | **99.4**% |

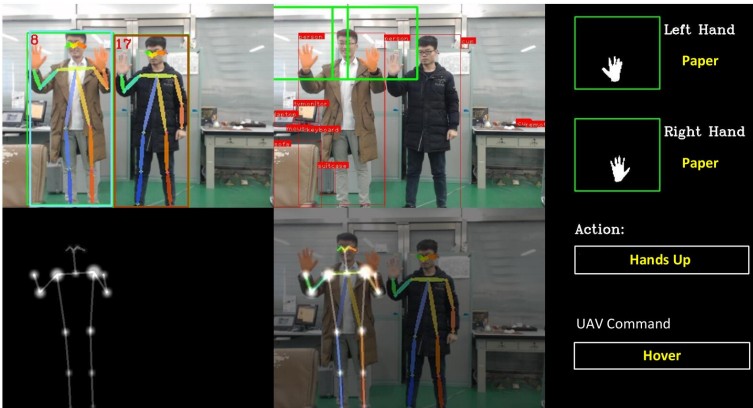

**Figure 23.** Simulation experimental result of our system when multiple person perform similar actions.

Finally, we applied our system to control the flight action of a UAV (as illustrated in Figure 19) under a real urban scene. In total, the flight experiments were approximately 3 h long, and approximately 324,000 images were tested under five types of flying conditions (buildings, factory, woods, indoors, and square). The total accuracy of final flight command sent to the UAV was over 99.5%. Figure 24 presents an example of our real flight experiment.

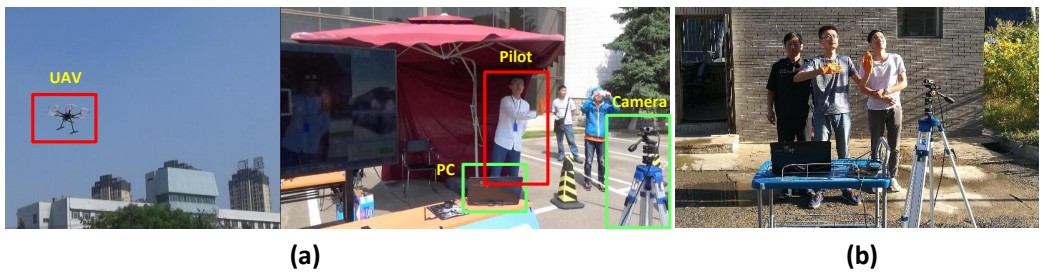

**Figure 24.** Experimental results of controlling the flight of a real UAV with the proposed system. (**a**) Under the real urban scene; (**b**) Under the strong illumination condition.

## 4. Discussion

As illustrated in Figure 14b, the false alarms of the action recognition were usually caused by similar actions. One solution to this problem is to select completely different actions as the features in the code book for different flight commands. Moreover, for the gesture recognition, Figure 25 indicates that false alarms arose in all of the compared algorithms. In our system, the false alarm was caused by the camera viewpoint, in which "Vertical Knife" resembled "Vertical Palm". The other compared algorithms usually produced false alarms on the background component or were affected by another person (or even parts of their own body).

According to Tables 5 and 6, the recall rate of our method changed significantly. Such variations in the system performance are caused by the noise in the dataset. As illustrated in Figure 26a,b, the depth image in the NYU dataset often contained other noise information such as the arm or wrist, and the segmented hand region ((c) and (d)) in our dataset was usually clear and easily recognized. This phenomenon is quite natural because better segmented images generally lead to improved recognition results. As for the accuracy loss in other algorithms like YOLO-V3, we consider that is caused due to the following reason: (1) less training samples (2000 samples) in our dataset (52,360 training samples in NYU dataset); (2) there is no background objects in NYU dataset (as shown in Figure 26a,b), but our dataset contains complex background conditions such as: indoor and outdoor, various background components and strong illumination changes (as shown in Figures 25 and 27).

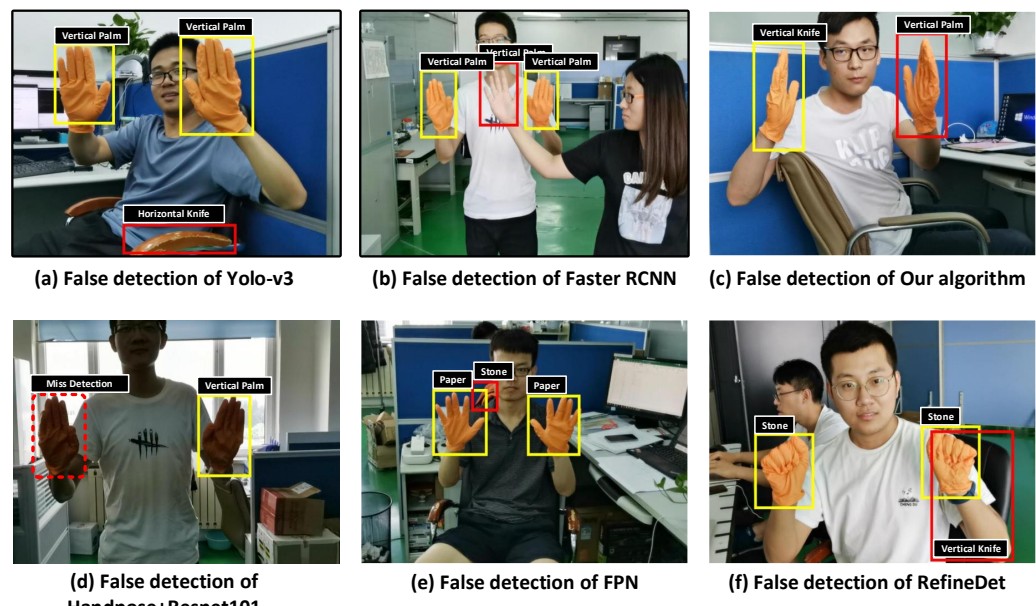

**Figure 25.** Examples of false alarms arised in all compared algorithms in our test dataset. The yellow rectangle refers to the correct recognition; the red one means the false alarm (solid line) or miss detection (dot line).

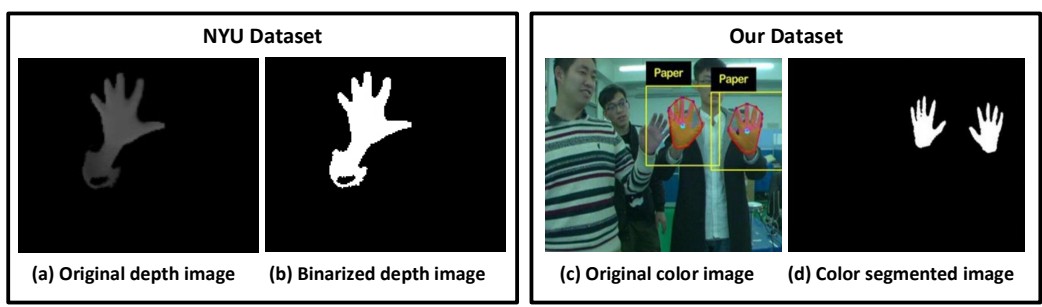

**Figure 26.** The noise in test dataset may degrade the performance of gesture recognition algorithms. (**a**,**b**) are the original depth image and corresponding binarized image in NYU dataset; (**c**,**d**) are the input image and segmented hand region in our dataset.

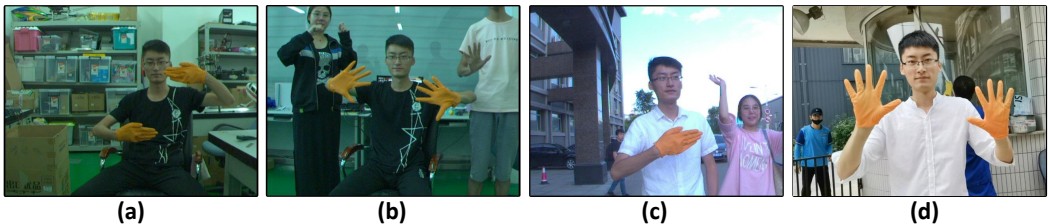

**Figure 27.** Complex background conditions of our dataset. (**a**–**d**) Including indoor, outdoor, multiple persons and cluttered background.

The processing speeds of each module in our system were as follows: operation module: 30 ms, action recognition module: 45 ms, gesture recognition module: 15 ms, scene understanding module: 2 ms, and cross-validation: 5 ms. Therefore, the total processing speed of our system was approximately 97 ms, and the signal delay between the PC and UAV was less than 0.1 s.

During our experiments on the real UAV platform, the working distance between the UAV and pilot was approximately 2 to 800 m (theoretically, a maximum of 2 km) with no occlusion. We tested

our UAV under three scenes, namely wood, urban, and factory, and the maximum wind speed was approximately 6.5 m/s during the test.

## 5. Conclusions and Future Works

We have proposed an intelligent human–UAV interaction system based on the joint cross-validation over action–gesture recognition and scene-understanding algorithm. The flight strategy of the UAV was obtained from the deep learning-based scene-understanding algorithm so that it could follow the flying conditions more smartly. Joint cross-validation was applied to the results of a multi-feature cascade gesture recognition algorithm and that of the AlphaPose+ST–GCN action recognition algorithm to improve the recognition accuracy of our system. The final flight command sent to the UAV was a combination of the flight action obtained from the cross-validation and the scene-understanding result. The effectiveness and efficiency of the proposed system were demonstrated by means of flight experiments on a real UAV with a total of 3 h of flying, and the overall accuracy of the translated flight commands was over 99.5%.

At present, the UAV pilot is requested to wear gloves, because these can aid our system in distinguishing the UAV pilot from surrounding persons (without gloves) who happen to perform similar actions to the pilot (for example, in the case illustrated in Figure 4), as well as improving the segmentation results of the hand regions. In the future, we would like to introduce a 3D RGB-D sensor (such as the well-known TOF camera Kinect V2) to replace the visible spectrum camera in our system, because it is more natural for a human to control the UAV without gloves. With the aid of 3D information, it will be easy to distinguish the UAV pilot (who is closest to the camera) from surrounding persons. To address some specific task such as UAV rescue/delivery, more important modules needs to be combined with our system such the onboard human detection module and mechanical delivery system to finish the rescue or or delivery operation. Before applying the proposed system to other intelligent robots such as industrial robots, self-driving vehicles, and aerial manipulators [52], we would like to perform further flight tests with additional users.

**Author Contributions:** C.H.: methodology ans software; B.C.: software and valdation, D.L.: validation, Y.H.: validation; J.H.: validation

**Funding:** This work is supported by the Program of National Natural Science Foundation of China (NSFC) under Award No. U1609210, No. 61573338, No. U1508208.

**Conflicts of Interest:** The authors declare no conflict of interest.

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
