# Peer review of "Intelligent Human–UAV Interaction System with Joint Cross-Validation over Action–Gesture Recognition and Scene Understanding"

_applsci, doi:10.3390/app9163277_

Round 1

Reviewer 1 Report

This paper describes a system which allows the user to control a UAV with combinations of gestures and actions by employing computer vision techniques. The UAV also adjusts its flight to the conditions of the environment (indoors/outdoors, over water, etc.).

I have several concerns about this work. The main one relates to the system's architecture and setup, which seems to be overall impractical for real-world utilization, by requiring a very complex system setting to perform a task which is very common these days: flying a UAV. Currently, there are many state-of-the-art systems which are equipped with technology that makes it possible for operators with little training to successfully control a UAV with a remote controller, and remote controllers are overall accepted by the community.

The motivation for the work is not convincing, with affirmations like "the interaction between UAV and human operator has never been changed", which can hardly be supported by references from 2014 and 2016, since there has been much work in this field in most recent years. A quick search on the topic immediately led me to find works like [1] and market solutions like [2], which not only disproves that the UAV-operator interaction has never been changed, but also disprove some other affirmations like "gesture (or action) recognition algorithms have never been applied in the human-UAV interaction for controlling the flight action of an UAV".

[1] Natarajan, Kathiravan, Truong-Huy D. Nguyen, and Mutlu Mete. "Hand gesture controlled drones: An open source library." 2018 1st International Conference on Data Intelligence and Security (ICDIS). IEEE, 2018.

[2] http://maestroglove.com/

Regarding the experiments, more details are required: description of the datasets collected by the authors (how many different individuals performed data collection? class distribution?); once there are user-dependent features extracted in the gesture recognition model, it is particularly important to understand with how many users was the system finally tested (did not find any reference to this matter); etc. Without this information, these experiments/results cannot support the conclusions of the paper.

Additionally, the authors should also refer to the applied safety and risk mitigation measures, especially when testing the system in a real-world scenario, like an urban area.

For these reasons, I believe that this paper should be rejected in its present form.

There are, however, some very interesting ideas and methods over the manuscript. My suggestion to the authors is that they take the focus out of the entire system and focus more on each of its modules:

1) The scene understanding algorithm, and its impact in adjusting the flight options of the UAV (regardless of how the UAV is being controlled; it can even be a remote controller, only the flight strategy will be adjusted)

2) The joint cross validation over Action-Gesture recognition, which seems to be a very interesting solution for improving the recognition performance of the system while enabling a broader range of UAV complex actions.

In this sense, in my opinion, the authors should focus in deepen their work in these 2 topics, separately, and, only then, when the approaches are mature enough, rethink the system design to include the combination of both modules while taking into consideration usability concerns (e.g. do not depend on using gloves; handle image light and contrast properly so that the operator does not need to be in the shade, like Fig. 21 indicates; etc.) and an extended validation (more flight time; tested with several users/operators; etc.).

Reviewer 2 Report

The paper is well written and the authors have presented their research in a well organised and clear fashion. Just a few suggestions for the others to consider:

A) Minor English grammar editing

     1) "a UAV" not "an UAV" - please note that the "a/an" indefinite article usage is based on pronunciation not just spelling, therefore "a UAV" and not "an UAV" is the correct usage because in the abbreviation the first letter 'u' is pronounced |yoo| a consonant. However, "an unmanned aerial vehicle" is correct because the 'u' in "unmanned" is vowel.

     2) "Figure 4" not "Fig ure4" in "Section 2.2 - Scene Understanding for UAV Controlling"

     3) Suggest writing Table references in full i.e. "Table 1" not "Tab.1" for easier reading and better clarity. Probably same with Figures if you want to be consistent i.e. "Figure 1" or "Fig. 1" (with a space before the figure number) and not "Fig.1". Also "Eq. 1" not "Eq.1"

     4) First paragraph on page 7 - 'which' not 'who' in "...represent the weight function who provides a weight vector..."

B) Additional content consideration

     1) What UAV platform was used? Or was this customly developed? In which case, what's the specification of the UAV?

     2) In the 5th paragraph of "Section 1 - Introduction", the authors suggested that "...gesture (or action) recognition algorithms have never been applied in the human-UAV interaction to controlling the flight action of an UAV", but that's not true because the commercial DJI Spark Drone released in 2017 has gesture recognition capability - which must rely on some gesture recognition algorithm. Surely, there exists some research examples too.

     3) Figure 3 suggested gesture was captured by a fixed camera on the ground/tripod. Could the camera on the UAV be used to capture the gesture instead?

     4) Also, it seems the scene and action gesture algorithms developed in this research were computed on a PC, could this have been processed on-board the UAV using a Linux-based single board computer such as the Odroid XU4, Raspberry PI 4, or Nvidia Jetson TX2?

C) Overall comment: 

     1) Generally, the images were clearly described and annotated for easy understanding. The authors description of their method such as scene understanding (adaptive flight strategy for changing scene) and operator module, were quite clear, and their results were well presented.

Reviewer 3 Report

The subject of the paper is interesting, well structured and useful, showing a quite convincing methodology. The novelty is not clear to the reader, due to the missing description of related works in the literature. The results seem promising but they are not clear, especially in the gesture recognition comparison.

INTRODUCTION:
- Figure 1-c is not adequate to its caption. UAV means Unmanned Aerial Vehicle, which is not the case of the vehicle in the image.
- This statement is made: "Although several reports have shown that it is possible to use the gesture (or action)-recognition-based methods for human machine interaction, gesture (or action) recognition algorithms have never been applied in the human-UAV interaction to controlling the flight action of an UAV[19].". Although it might be supported by the citation, it was made in 2015  and is not true anymore. The authors must revise the literature and provide a background of the works with similar applications.

PROPOSED SYSTEM:
- Figure 6 and Table 1 present two distinct meaning of the word "Strategy". Please consider replacing "Strategy" by "Scene" in the figure.

EXPERIMENTAL RESULTS:
- Some knowledge about the cameras' resolution would be useful to better understand the algorithm's working conditions.
- Table 4 caption says 5.2k training images, but the text beneath says 52k.
- In section 3.2, the difference between Recall Rate of the proposed algorithm in Tables 5 and 6 (too high in the last), may suggest that the algorithm is somehow dataset-dependent or that the training procedure resulted in a dataset memorization. The reason gave in the text is not clear.
- This issue is raised in the Introduction section: "... when delivering packages from a start point α to destination β that is out of vision from α, instead of waiting for the command transmitted from its operator at point α, an UAV should be more smart to be able to directly co-operate with the receiver at the destination point β. That is because the time delay caused by transmitting signals at long distance is too long for the UAV to deal with the abrupt conditions[6].". However, it is not discussed in the results. When it is said that the the signal latency is less than 100ms, the working conditions shall be described aswell (distance to the UAV and if there are occlusions) because the control of the UAV might be compromised.

DISCUSSION AND FUTURE WORKS:
- No discussion is made here, there are only presented future work suggestions. The discussion exists, but is being made along the Experimental Results section. Maybe a discussion of the combined system behavior would increase the paper quality.

GENERAL:
- It is advisable to recheck the English language and style in the text and images.
- There are some inconsistencies between the text and the tables/figures. For example, Table 2 has 4 "undefined" actions, but the text says 6.
- The formatting of the references to Tables and Figures shall also be revised.

Round 2

Reviewer 1 Report

Thank you to the authors for taking into consideration my comments.

Introduction / Proposed system

I appreciate the authors for fairly extending the background/related work. I believe the authors have verified that there is a lot of work developed in this area, and, so, for this system as a whole to be considered innovative, it must present clear advantages over what already exists. However, the motivation (Sections 1 and 2) for conceiving this system is still inconsistent.

1)

"However, in the case of UAV rescue operation as shown in Figure 2 (c), human-UAV interaction is far beyond flying a UAV. Under such condition, since the person needing rescue could not contain a remote controller, the UAV needs to understand the human action (or gesture) by itsself but not by the UAV pilot (especially in the case of teleoperation, the time delay between UAV and its pilot is critical important)." (3rd paragraph, Section 1)

"Such ability is also important for a UAV to finish many difficult tasks. For example, like Figure 3 (c), (d) (where the person needs to be rescued
or package receiver does not contain a remote controller), the gesture/action-based human-UAV interaction will play an important role in the rescue operation (or delivery task). We do belive that such interaction method is not limited to UAV but with the potential application for all kinds of robots (like industrial robot, unmmaned taxi as shown in Figure 3 (a), (b))"
(2nd paragraph, Section 2)

The person being rescued is not flying the UAV - an external operator is. So, why can't the external operator use a remote controller?

As far as I understand, the system is not ready for receiving commands from the person being rescued. That person does not know how to control the UAV, may be in circumstances under which it is not possible to control it (e.g. underwater), etc. Moreover, they do not have the gloves, nor the fixed video camera required by the system.

So, in which way does your system provide better assistance for rescue operations or even delivery tasks, if there is no interaction planed by your system with the dropping/target point/location?

2)

I believe the following argument would be enough to justify the research design: "In this paper, we proposed a gesture-action joint recognition method of human-robot interaction and tested it on the real UAV platform."

If standing solely with this argument, the background research, however, should not focus on the state-of-the-art of human-UAV interaction, but human-robot interaction, in a broader way.

However, if the authors persist in arguing that this system brings significant advantages to solve problems of UAV control, they should present sustained evidence of those advantages.

Discussion of results

The works mentioned on the Introduction ("Other researchers also tried to apply the recent deep-learning-based gesture/action algorithms (such as LeapMotion [2],[44], OpenPose[45], LSTM[47], YOLO[46] and P-CNN[47]) into human-UAV interaction. The general recognition accuracy of these works varies from 90% to 92% which is not good enough for controlling a UAV.") must be compared with the approach of this work in the discussion section.

In which way is your work compared to theirs? Are the study methodologies and performance comparable? With how many users (and for how many hours) did they test their system?

Only by having this discussion can the authors sustain affirmations like "The effectiveness and efficiency of the proposed system has been proven through the flight experiments on a real UAV of totally 3-hours flying and the total accuracy of translated flight command is over 99.5%." and claim contribution 2 - if the system proves to be better than those which already exist.

Final remarks

As my comments suggest, I am still not convinced that this system is necessary as a whole, and would like to reinforce the suggestion of my previous review: the authors should focus in validating the quality of the developed modules separately instead of the entire system. This suggestion was not addressed by the authors in their response to the comments.

Reviewer 3 Report

First of all, I would like to thank you for taking into consideration my comments on the revised version. 

In what concerns the methodology, I have just one more comment related to this answer:
"Reply: Such difference between the recall rates of two datasets is caused by the variation of segmentation results. In NYU dataset, the segmented hand regions usually contain parts of arms and the human fingers are not well separated. Since all the gesture recognition algorithms are applied to the segmented hand regions, the performance of all algorithms in NYU is worse than that in our dataset."
I can agree with this statement up to some point. A significant improvement can only be seen in Our algorithm and FPN. In the other methods, the improvement is not that notorius, existing even a loss of accuracy in the Yolo-v3
This loss might be related to the fact of your dataset having a significantly lower number of train samples, when compared to the public dataset. Although, the proposed algorithm is better than the other in both cases, and that is mentioned on page 15. The only part that is confusing is when is stated that "The main reason for the difference between the results of our dataset and public dataset is that the background of our dataset is more complicated than that of public dataset, where hand-like objects in the background are easy to be recognized as hand." 
What do you mean by "more complicated"...?
OTHER COMMENTS:
- Page 1 - cite3 is not well-formatted;
- Figure 18 - I think that the term "virtual" would be more appropriated than "electronic".
LANGUAGE AND STYLE:
- Again, I strongly recommend you to re-check the English-style with an expert or, at least, use some spell and grammar checker. There are quite some English errors, especially in the newly written part.
For example, I can somehow understand the content of this statement, but it doesn't make sense as it is:
"Under such condition, since the person needing rescue could not contain a remote controller, the UAV needs to understand the human action (or gesture) by itsself but not by the UAV pilot (especially in the case of teleoperation, the time delay between UAV and its pilot is critical  important)."
